# Hybrid Carbon Nanotubes–Graphene Nanostructures: Modeling, Formation, Characterization

**DOI:** 10.3390/nano12162812

**Published:** 2022-08-16

**Authors:** Alexander Yu. Gerasimenko, Artem V. Kuksin, Yury P. Shaman, Evgeny P. Kitsyuk, Yulia O. Fedorova, Denis T. Murashko, Artemiy A. Shamanaev, Elena M. Eganova, Artem V. Sysa, Mikhail S. Savelyev, Dmitry V. Telyshev, Alexander A. Pavlov, Olga E. Glukhova

**Affiliations:** 1Institute of Biomedical Systems, National Research University of Electronic Technology MIET, Shokin Square 1, 124498 Moscow, Russia; 2Institute for Bionic Technologies and Engineering, I.M. Sechenov First Moscow State Medical University, Bolshaya Pirogovskaya Street 2-4, 119991 Moscow, Russia; 3Scientific-Manufacturing Complex “Technological Centre”, Shokin Square 1, bld. 7 off. 7237, 124498 Moscow, Russia; 4Institute of Nanotechnology of Microelectronics of the Russian Academy of Sciences, Leninsky Prospekt 32A, 119991 Moscow, Russia; 5Institute for Regenerative Medicine, I.M. Sechenov First Moscow State Medical University, Bolshaya Pirogovskaya Street 2-4, 119991 Moscow, Russia; 6Department of Physics, Saratov State University, Astrakhanskaya Street 83, 410012 Saratov, Russia

**Keywords:** carbon nanomaterials, carbon nanotubes, graphene oxide, hybrid nanostructures, laser formation, field emission, cathode, adhesion, welding, mechanical properties, defects

## Abstract

A technology for the formation and bonding with a substrate of hybrid carbon nanostructures from single-walled carbon nanotubes (SWCNT) and reduced graphene oxide (rGO) by laser radiation is proposed. Molecular dynamics modeling by the real-time time-dependent density functional tight-binding (TD-DFTB) method made it possible to reveal the mechanism of field emission centers formation in carbon nanostructures layers. Laser radiation stimulates the formation of graphene-nanotube covalent contacts and also induces a dipole moment of hybrid nanostructures, which ensures their orientation along the force lines of the radiation field. The main mechanical and emission characteristics of the formed hybrid nanostructures were determined. By Raman spectroscopy, the effect of laser radiation energy on the defectiveness of all types of layers formed from nanostructures was determined. Laser exposure increased the hardness of all samples more than twice. Maximum hardness was obtained for hybrid nanostructure with a buffer layer (bl) of rGO and the main layer of SWCNT—rGO(bl)-SWCNT and was 54.4 GPa. In addition, the adhesion of rGO to the substrate and electron transport between the substrate and rGO(bl)-SWCNT increased. The rGO(bl)-SWCNT cathode with an area of ~1 mm^2^ showed a field emission current density of 562 mA/cm^2^ and stability for 9 h at a current of 1 mA. The developed technology for the formation of hybrid nanostructures can be used both to create high-performance and stable field emission cathodes and in other applications where nanomaterials coating with good adhesion, strength, and electrical conductivity is required.

## 1. Introduction

The cathode is the main element of systems generating electron flows. The key performance indicators of cathodes are the values of emission current and threshold voltage, as well as stability during long-term operation. Thermionic emission devices demonstrate the highest emission current values at the low threshold voltage and time stability. Field emission is an actively developing area, which, because of the principle of operation, is resistant to fluctuations in the temperature of the medium, does not require warm-up time, and has a monochromatic electron energy spectrum. However, the problem with the time current stability has not been solved at present [1]. A wide range of materials have been studied as a field emission cathode material: nanowires (GaN [2], ZnO [3], GaAs [4]), nanoparticles (In_2_O_3_ [5], GeSn [6], InP [7]), one-dimensional materials—carbon nanotubes (CNT) [8,9,10,11], two-dimensional materials (graphene [12,13,14], MoS_2_ [15]). Carbon nanomaterials have outstanding characteristics of high strength, good electrical and thermal conductivity, and low work function and are the most promising materials for the formation of field emission cathodes [16].

CNT have outstanding emission properties due to their excellent electrical conductivity and high aspect ratio. Together with the unique thermal and mechanical properties, this makes it possible to use them as additives to improve the characteristics of piezoresistive elements [17,18], solar energy converters [19], conductive and biocompatible polymers [20,21,22], and energy storage devices [23]. One of the promising applications of CNT is the creation of electronic devices with CNT cathodes as electron sources with a low threshold voltage of field emission cathodes [24,25,26]. Graphene and its derivatives are also widely used in the creation of electronic devices. High values of a specific area and high electrical conductivity make it possible to create supercapacitors [27,28,29,30,31], sensors [32,33,34], and other devices based on them [35,36,37]. The high electrical conductivity of graphene gives excellent shielding properties to composites based on it [38]. In combination with biopolymers, the toxicity of graphene can decrease, so graphene-based biocomposites and biosensors can be created [39]. Graphene structures are also widely used in the creation of electronic emitters [40,41,42]. Composites based on rGO have great potential for creating field emitters [43]. In the case of both CNT and graphene, the emission is provided by the presence of atoms at the edges of CNTs and graphene flakes that form a distorted sp^3^-hybridized geometry instead of a planar sp^2^-hybridized configuration. As a result, localized states are formed at the edges, and, consequently, potential barriers to electron emission are reduced [44].

The first description of emission from individual CNTs is known, where the emission current was from 0.1 to 1 μA/cm^2^ at a bias voltage of less than 80 V [45]. One of the first works describing CNT film cathodes showed that the field emission current density was about 0.1 mA/cm^2^ at an applied voltage of 200 V and 100 mA/cm^2^ at 700 V [46]. In most works describing CNT-based emission cathodes, vertical arrays of CNT grown by the chemical vapor deposition (CVD) method on substrates are used as an electron source [47,48,49]. The advantages of such vertically grown CNT arrays are a high degree of adhesion to the substrate and low-resistance contacts between nanotubes and substrate [50]. However, because of the high temperature and active environment of the CVD process, a limited range of substrate materials can be used. Additionally, the features of this method for vertical arrays of CNTs synthesis include significant resource costs and complexity in implementation [51]. In this regard, more accessible transfer methods for creating emission cathodes based on disordered films of carbon nanomaterials are being actively developed [52]. Point emitters with films of randomly oriented CNTs can demonstrate high current densities up to 105 A/cm^2^ [53]. Such point emitters provide a localized beam of electrons and are applicable in the creation of microwave amplifiers and microfocus X-ray sources [54,55]. CNT film emitters can show stable uniform emissions from a large area with a current density of several mA/cm^2^ [56,57]. Graphene-based emitters are also capable of delivering current densities of several mA/cm^2^ [58,59].

Significant progress has been achieved in the creation of emission materials based on CNT and graphene/graphene oxide. These two modifications of nanocarbon can advantageously complement each other’s structural and electrophysical properties, which makes it possible to demonstrate emission current values that exceed those for CNT and graphene separately [60,61]. When creating such hybrid CNT/graphene nanostructures, the methods of growing CNT on graphene flakes [62,63] and growing graphene on CNT arrays [60] are mainly used. When CNT/graphene nanostructures are grown by various methods, stable bonds of CNT with graphene are formed [64,65,66]. However, the high cost and complexity of such methods do not allow producing efficient CNT/graphene emitters on an industrial scale. Another disadvantage of CNT/graphene growth methods, as it was mentioned above, is a limited range of materials that can be used as substrates. More affordable film deposition methods are used to create CNT/graphene emission films, including electrophoretic deposition [67], plasma deposition [68], filtration with optical influence [69], vacuum filtration [70], drying and reduction [71]. When synthesizing emission structures based on graphene, Microwave Plasma Enhanced CVD can be used [58]. When synthesizing emission structures based on graphene oxide, the Hummers method can be used [59]. Both methods involve the use of high temperatures, which significantly limits the choice of substrates, especially for biomedicine [21,72]. In addition, when synthesizing carbon nanomaterials on substrates, it is difficult to select the synthesis parameters that critically affect the characteristics of the obtained nanomaterials. The use of studied and commercially available carbon nanomaterials with their subsequent spray deposition on arbitrary substrates makes it possible to select the parameters of nanomaterials after their synthesis. SWCNT can be separated according to the conductivity type, which will allow the formatting of effective structures for various applications based on semiconducting or metallic SWCNT [73,74].

Spray deposition is an effective and extremely simple film deposition method for creating films from carbon nanomaterials. Multiwalled CNT films deposited by this method can demonstrate high field emission current density of 13 mA/cm^2^ [75]. However, the key parameter of the field emission of cathodes, the stability of the emission over time, was confirmed. During spray deposition of dispersions based on SWCNT, an emission current density of 60 µA/cm^2^ was achieved. However, in this case, the stability of the current over time was verified. The stability of emission from carbon nanomaterials over time is positively affected by the addition of a polymer to the composition of the sprayed dispersion [76]. In this case, high stability of emission over time can be achieved, but at a low emission current density of 1 mA/cm^2^. Thus, when creating field emission cathodes by the spray deposition method, there is a need to further improve their emission properties. This can be achieved both by improving the structural characteristics of carbon nanomaterials with external influences and by searching for an advantageous combination of different carbon nanomaterials.

To improve the emission properties of materials based on nanocarbon modifications, various methods of structural modification are used. There are a large number of chemical methods for structuring carbon nanomaterials [77,78,79]. However, preference is given to electromagnetic methods because of their availability and simplicity. Directed electromagnetic radiation makes it possible to structure nanotubes by increasing the angle of their orientation relative to the substrate [80]. Laser exposure makes it possible to change the structure and shape of nanomaterials by localizing high-energy electromagnetic radiation on the surface of nanomaterials [81,82,83,84,85,86]. The laser beam positioning system allows setting the required geometry of the laser exposure area. During exposure, laser energy is absorbed by electrons and transferred to CNT atoms. In its turn, the appearance of high-energy phonons leads to the formation of defects such as vacancies and interstices in the walls of nanotubes. This stimulates the formation of new bonds on the contact surfaces of the nanotubes.

Previously, it was demonstrated that under high-intensity laser exposure with an energy density of 0.3–0.5 J/cm^2^ on nanomaterials containing single-walled carbon nanotubes (SWCNT), multiwalled carbon nanotubes (MWCNT), and rGO flakes to nanotubes and nanotubes with each other are welded forming branched CNT/rGO networks. Contacts were formed between the regions at the rGO flakes ends and the defective side surfaces and ends of the nanotubes. In this case, the maximum electrical conductivity of 22.6 kS/m was achieved for the MWCNT/rGO hybrid nanostructure [87].

In addition to improving the emission properties of carbon nanomaterials by external action, it is necessary to take into account the presence of contact resistance between the emitting layer and the substrate. The contact resistance of the CNT substrate has a significant effect on the field emission efficiency in CNT cathodes. In the field emission process, electrons tunnel through two potential barriers: first, the electrons cross the barrier between the CNT and the substrate, and then they emit into the vacuum from the CNT. A buffer layer is often used to reduce the contact resistance between the CNTs and substrate. The material of such a layer can be made of metals, for example, silver [88]. Materials such as Ti, TiN, W, and Mo are often used as a sublayer, and Ni, Co, and Fe—as catalysts for the CNT synthesis. In addition to participating in the chemical process of CNT synthesis, they are able to reduce the contact resistance between CNT and the substrate [50,89,90]. The presence of a buffer layer can increase the maximum field emission current; for example, in the article [91], the achievement of a stable current density of 30 mA/cm^2^ is shown. Thus, buffer layer formation before the main cathode structure formation can solve the main problem of field emission cathodes—the current time stability. To ensure the best match between the buffer layer morphology and the field emission structure itself, it is most advantageous to use a buffer layer that consists directly of carbon nanomaterials and can provide high adhesion to the substrate. Thus, in work [92], the authors used the welding of carbon nanotubes to a silicon substrate using continuous laser radiation. Based on the above, there is a need for a universal method for creating hybrids of carbon nanomaterials, which makes it possible to control their parameters.

This paper presents the results of the formation of the nanostructures based on SWCNT, rGO, and hybrids of SWCNT and rGO on silicon substrates using pulsed laser radiation. It was demonstrated that different types of carbon nanomaterials could complement each other because of their different morphologies. Thus, hybrid nanostructures with excellent mechanical and electrical properties can be created. When using labor-intensive synthesis methods, it is difficult to simultaneously control the complex parameters of the resulting nanomaterials. The main advantage of the created hybrid nanostructures using SWCNT and rGO laser welding is the possibility of using commercially available and well-studied carbon nanomaterials with specified parameters. In addition, the list of materials that can be used as substrates for the formation of nanostructures using laser radiation is expanding. Using molecular dynamics modeling and electron microscopy, laser induction of rotational moment in graphene flakes is shown. The effect of the rGO buffer layer on improving the mechanical properties of hybrid nanostructures and their field emission, including the stability of emission in time, were also demonstrated.

## 2. Materials and Methods

### 2.1. Method for Modeling the Carbon Nanomaterials Behavior under the Influence of Electromagnetic Field of Laser Radiation

Self-consistent density functional tight-binding (SCC-DFTB) and real-time time-dependent density functional tight-binding (RT TD-DFTB) implemented in the DFTB+ Version 20.2 software package (Free Software Foundation Ltd., Boston, MA, USA) methods were used to build atomistic models of layers from graphene flakes and hybrids of nanotubes with graphene flakes, as well as to simulate the response of the atomic and electronic structure of layers to laser radiation [93]. Searching for an energetically favorable configuration and size of the atomic structure of supercells was determined through preliminary optimization, which comprised varying coordinates and dimensions of all atoms of the supercell in order to achieve a global minimum of the total energy.

The response to laser action was simulated using the RT TD-DFTB method, which implements Ehrenfest quantum semiclassical molecular dynamics. The solution of the wave equation for electrons is
(1)$$jћ\frac{\partial \psi }{\partial t}=H_{e}\left(r_{i},R_{I}\left(t\right)\right)\cdot \psi \left(r_{i},R_{I}\left(t\right)\right)$$
together with the classical equation of nuclei motion
(2)$$M_{I}\ddot{R_{I}}\left(t\right)=-\nabla_{I}\int dr\;\psi^{*}H_{e}\psi .$$

In Equations (1) and (2) $H_{e}\left(r_{i},R_{I}\left(t\right)\right)$—electronic part of Hamiltonian, determined by electron coordinates $r_{i}$ and coordinates of the nuclei $R_{I}$, changing over time, j—imaginary unit, $M_{I}$—nucleus mass with number I, ψ—electron wave function, which is written as a linear combination of atomic orbitals
(3)$$\psi \left(r;t\right)=\underset{k}{\sum }a_{k}\left(t\right)\varphi_{k}\left(r;R\left(t\right)\right).$$

The fundamental difference between Ehrenfest’s molecular dynamics lies in the time dependence of the weight coefficients $a_{k}\left(t\right)$. Nuclei move in accordance with classical mechanics in the effective potential $V_{e}^{E}$ created by electrons. Here E indicates that the potential field is created by electrons at a specific value E (one of the possible energy values). This potential is calculated as the average value of the electronic part of the Hamiltonian at the coordinates of the nuclei $R_{I}\left(t\right)$, fixed at the moment t (in the framework of Self-consistent field (SCF)—mean field theory) [94,95].

The laser field is
(4)$$F\left(t\right)=F_{0}\;f\left(t\right)\;sin\left(\omega t+\varphi \right),$$
where ω—laser pulse frequency, φ—phase, $F_{0}$—maximum value of laser field, the function $f\left(t\right)$ determines the shape of the laser pulse. In this work, the Gaussian form is used in the modeling, as in the experimental part:(5)$$f\left(t\right)=exp\left\{-\frac{{\left(t-t_{m}\right)}^{2}}{\beta^{2}}\right\},\;\beta =\frac{\tau }{2\sqrt{\pi }},\;$$
where $t_{m}$ is the time at which the pulse is centered, and τ is the duration of the pulse.

### 2.2. Method for Creating Carbon Nanomaterials Dispersions and Applying Them to Substrate

To form layers of carbon nanomaterials, it was necessary to prepare liquid dispersed media. SWCNT Tuball (OCSiAl Ltd., Moscow, Russia) was synthesized by the gas phase method. The SWCNT diameter was 1–2.5 nm, the length was about 5 µm, and the specific surface was 420 m^2^/g. rGO was synthesized using a modified Hummers method. For this, 1 g of graphite powder was mixed with 6 g of potassium permanganate KMnO_4,_ and 14 mL of 85% phosphoric acid H_3_PO_4_ was added to the mixture. The whole mixture was mixed with 120 mL of 95% sulfuric acid H_2_SO_4_. The resulting dispersion was settled for 12 h at a temperature of 50–60 °C. Further, the dispersion was mixed with 140 mL of water, and 30% hydrogen peroxide H_2_O_2_ of ~20 mL was added to the resulting aqueous dispersion until foaming ceased. Deionized water with a resistance of at least 16.5 MΩ × cm was used. Then, the sediment was separated from the dispersion by centrifugation using an Avanti J-30I centrifuge with JA-30.50Ti rotor (Beckman Coulter Inc., Brea, CA, USA) at 12,000× *g* for 30 min. Next, the remaining dispersion without sediment was washed with water to pH 5. The reduction of graphene oxide was carried out by annealing in a muffle furnace at a temperature of 200 °C for 1 h and subsequent annealing in the argon and hydrogen atmosphere at 1000 °C with a volume ratio of 1:1 for 1 h. The number of rGO graphene layers did not exceed 4. It was found that rGO contained C–H bonds in its structure [96].

Preparing dispersions for the formation of layers on a substrate was carried out by mixing carbon nanomaterials with solvent. Dimethylformamide (DMF) was used as a solvent. The dispersion compositions were as follows: 1 dispersion—SWCNT with concentration 0.1 mg/mL, 2 dispersion—rGO with concentration 0.1 mg/mL, 3 dispersion—rGO + SWCNT with weight ratio 1/1 and with concentrations 0.05/0.05 mg/mL. Table 1 presents the dispersion compositions. After mixing, the dispersions were processed with the immersion ultrasonicator Q700 Sonicator (Qsonica Ltd., Newtown, CT, USA) within 10 min at a power of 150 W/cm^2^. The dispersions were then subjected to ultrasonic treatment in the bath Elmasonic S30H (Elma Ltd., Singen, Germany) (power 80 W) within 60 min. Separation of dispersions was carried out using the centrifuge at 20,000× *g* for 30 min at 15 °C. After centrifugation, the sediment was removed, and the remainder of the resulting dispersion was taken from each tube.

Thin films of carbon materials from dispersions 1, 2, and 3 were deposited by the spray deposition method. Substrates of heavily doped silicon (Si substrate), divided into chips 7 × 7 mm in size, which had a high thermal conductivity and good electrical conductivity, were chosen as substrates. Then substrates were laundered in the standard manner in Piranha, washed with deionized water, and dried in isopropyl vapor. For film deposition, the E2V dispensing system (Nordson EFD, Westlake, OH, USA) was used, which was a spray module mounted on a three-coordinate positioning system. The pressure for the air flow supply was 20 bar. The pressure for dispersions supplying was 0.05 bar. The nozzle diameter was 0.5 mm. The substrates were mounted on a heating stage for accelerated evaporation of the solvent from the formed layer. The stage was heated to a temperature of 120 °C. The layer formation with a thickness of ~500 ± 100 nm was carried out in 800–1000 passes, depending on the type of carbon nanomaterial. The indicated deviation of ±100 nm was the maximum deviation of the layer thickness, which was equivalent to 3σ. The standard deviation (σ) did not exceed 30 nm. The spread in thickness was due to the peculiarity of applying carbon nanomaterials by the spray deposition method, which was made by means of multiple passes of the nozzle over the sample. The duration of the layer deposition process could reach 1 h. During this time, partial decantation of the carbon nanomaterials dispersion took place. As a result, inhomogeneity was formed over the layer thickness.

For research, five groups of film samples were initially obtained from dispersions 1, 2, and 3. Each group contained 10 samples to get statistical results in studies.

### 2.3. Method of Laser Formation of Hybrid Nanostructures

The obtained samples of Si substrates with deposited layers of carbon nanomaterials were subjected to laser exposure. A laser setup was used, the main element of which was a Yb laser. The laser generated radiation at the main harmonic in the IR range of 1064 nm and operated in a pulsed mode with a pulse duration of 100 ns. The laser pulsed radiation mode is necessary for the local transfer of high energy in a short period of time to achieve the effect of nanowelding because it is known that short-duration pulsed radiation generates nonlinear optical effects in carbon nanotubes and graphene [97,98]. Since problems of emission electronics have demanded forming cathodes with a given area in certain regions of the topology, the laser radiation is directed using an accurate galvanometric scanning system along the X and Y coordinates (on the area). It is known that changing the pulse frequency of a laser beam scanning the sample also changes the spot overlap of consecutive laser pulses [99]. Thus, to achieve a uniform laser exposure of the carbon nanomaterials film area, the frequency of laser pulses was selected. At a beam positioning speed of 500 mm/s and laser beam diameter of 35 µm, a laser pulse frequency of 30 kHz was chosen. As a result, the laser pulses were superimposed on each other with a 17 μm overlap. This provided a uniform exposure of the film’s given area. The size of the formed treated area was 3 × 3 mm. The beam was focused using an object lens that provided a diffraction length greater than the film thickness. The laser beam profile had a Gaussian shape.

For sample rGO(bl)-SWCNT, the rGO layer was deposited and then exposed to laser radiation (0.8 J/cm^2^). It allowed the creation of a buffer layer that provides a high degree of adhesion of subsequent layers of carbon nanomaterials to the substrate. After that, the main layer of SWCNT was deposited and exposed to laser radiation (0.5 J/cm^2^).

To eliminate the influence of the atmosphere on the morphology of carbon nanomaterials layers, processed by laser radiation, two approaches were used. In the first case, a chamber with an Ar inert gas was used [100]. In the second case, a vacuum chamber with quartz glass was used, where the residual pressure was 0.1 mbar. A schematic representation of the setup for laser exposure of samples using a vacuum chamber is shown in Figure 1. The laser exposure on the specified areas was carried out according to the scheme described below. One of the square areas was left untouched as a reference sample. All samples from three groups were subjected to this effect.

### 2.4. Scanning Electron Microscopy

The study of the carbon nanomaterials structure on Si substrates was carried out using Helios G4 (FEI Ltd., Hillsboro, OR, USA) scanning electron microscope. The accelerating voltage of the electron column was 5 kV, and the electron probe current was 50 pA. The pressure in the vacuum chamber was 3.9 × 10^−4^ Pa.

### 2.5. Raman Spectroscopy

Raman Spectroscopy is an efficient nondestructive tool for analyzing the morphology of various nanomaterials, especially carbon nanostructures such as carbon nanotubes, graphene, etc. [101]. Based on the data on the intensity and frequency of Raman active vibrational modes, one can draw conclusions about the purity of the material, the presence, and types of defects, and also analyze the functionalization by various groups [102,103].

Raman measurements of carbon nanomaterial layers on Si substrate were obtained using inVia Qontor (Renishaw plc, New Mills, UK) in the backscattering geometry. Spectra were collected using 1200 1/mm grating and a semiconductor laser with a wavelength of 532 nm as an excitation source. Focusing of the laser beam on the sample surface was performed using a built-in microscope with a ×50 objective. The microscope output power was 0.6 mW. Each sample was measured three times (at a random location on the sample surface) to obtain statistically correct data. Calibration was performed before the measurement of each sample by recording Raman spectra of single crystal silicon.

### 2.6. Mechanical Characteristics

The stability of structures is important for the field emission cathode’s formation, as well as for studying the effect of welding structures. Thus, the hardness of the formed carbon nanomaterials nanostructures on Si substrates was measured before and after laser exposure. For measurements, a NanoScan-4D Compact nanohardness tester (TISNUM Ltd., Moscow, Russia) with a Berkovich pyramid-shaped indenting tip was used. The sample hardness was measured by smoothly immersing the indenter to a depth of 200 nm. To prevent imprints from overlapping, the distance between the measurement points was 100 µm. The time to reach the required depth (Load time) was 10 s, and the time to hold the achieved load (Hold time) was 1 s. For each sample, five measurements were carried out, then the average hardness values were calculated.

The degree of nanostructures adhesion was assessed by the sclerometer method. This method consisted in applying a scratch to the film surface to establish the force with which the coating will be detached from the substrate. For the experiment, the above-described nanohardness tester was used. Scratching was carried out by smoothly immersing the indenter with a varying load from 10 to 60 mN. The length of the applied scratch was 500 μm. The tip speed was 100 nm/s, the load time was 10 s, and the hold time was 1 s.

### 2.7. Field Emission Characteristics

The study of field emission homogeneity over the samples surfaces was carried out in a vacuum chamber at a pressure of at least 1 × 10^−6^ mbar using the MCS-3D positioning system based on linear stages SLC-17 (SmarAct GmbH, Oldenburg, Germany) with an accuracy at least 50 nm along the X, Y, and Z axes. To record the current characteristics, an anode with a curvature radius of 350 µm was installed on the positioning system. Figure 2 shows a schematic representation and external view of the experimental setup.

Evaluation of emission homogeneity for each sample was carried out for at least five points. The anode was placed above the sample surface. Further, the anode smoothly moved down along the Z-axis. When the anode was moving, the resistance between the anode and cathode was controlled. The anode shifting continued until a value of less than 10 MΩ was obtained. This point was taken as 0 along the Z-axis. Then the anode was moved up by 5 μm, and the current–voltage characteristic was measured.

To measure the integral field emission, a highly doped 7 × 7 mm^2^ silicon chip was used as an anode. The cathode was a highly doped 7 × 7 mm^2^ silicon chip with a deposited layer of selected carbon nanomaterials. The layer was formed through a metal mask with a 1.1 mm round hole. A 10 × 10 mm^2^ mica plate was used as a dielectric between the cathode and anode. A plate was 100 µm thick, with a 3 mm round hole in the center. A 2410C (Keithley Inc, Cleveland, OH, USA) high-voltage source-measurement device was used to measure field emission values.

## 3. Results

### 3.1. Modeling of Laser Formation of Hybrid Nanostructures from Carbon Nanomaterials

Initially, the influence of laser pulses on the atomic network of the layer surface, which was a diverse combination of graphene flakes, was established. For this purpose, a supercell of a thin 2D layer of ~4.2 nm thick (size along the Z-axis) containing 3874 atoms was built. The dimensions of the periodic box X × Y were 4.83 and 4.13 nm, respectively.

Figure 3a shows a general view of a layer fragment. Atoms of one supercell, which are formed by several supercells, are shown in blue. Figure 3b,c shows two types of supercells, demonstrating the features of the atomic structure of the constructed model of the quasi-2D layer. One view shows the topological structure and the presence of various graphene fragments, including flakes of different shapes, covalently attached to each other (Figure 3b). Another view shows the presence of nonhexagonal elements: five in red, seven in blue, and eight in green squares (Figure 3c).

The constructed atomistic supercell model included 24 octagons, 75 pentagons, and 72 heptagons. The surface of the film was represented by clearly visible graphene flakes, which formed a relief that was most suitable for the field emission of the cathode since each nanoflake can act as an electron nanoemitter in strong electric fields.

The modeling of the laser pulse action implied using a 1064 nm laser wavelength. The pulse duration (τ) varied from several tens of femtoseconds to several hundred (the minimum value was 30 fs, and the maximum was 250 fs). First, the detachment process of one carbon atom, which was the edge atom of a graphene nanoflake and had only two bonds with neighboring atoms, was investigated. The intensity of F_0_ was 0.001, 0.01, and 0.1 V/Å. The direction of the intensity vector was set as (0,0,−1). Thus, a large series of molecular dynamics studies was carried out with different energy values (E_0_) and τ, while in each case, the energy characteristics were recorded in increments of 0.01 fs: total energy, non-SCC energy, SCC energy, spin energy, external field energy, repulsive energy, nuclear kinetic energy, and dispersion energy.

As a result, it was found that, regardless of the value of F_0_ and τ, the detachment of the edge atom from the moment the laser pulse began already occurred at ~15 fs; at least after ~15 fs, the C–C bond length increased to ~1.8 Å, and then it increased until the complete detachment of the atom from the initial graphene structure. Figure 3d shows the profile of the kinetic energy of the nuclei (the curve is highlighted in green) during the first 40 fs at τ = 30 fs. The shape of the laser pulse is also shown in red in this figure. The kinetic energy of nuclei is one of the most important energy characteristics of the interaction process of the atomic system with laser radiation since it very accurately reflects the process of transferring the energy absorbed by electrons to the nuclei.

It can be seen that the kinetic energy of the nuclei profile was characterized by five local maxima over the indicated time interval. It should be noted that a similar profile was observed for all variations of the molecular dynamics modeling. This indicated the truth of the detachment process of the carbon atom established in the field of in silico research. Thus, it was found that when the second local maximum was reached, the mechanism of carbon atom detachment was “launched”. This happened at ~13th fs. The third local maximum indicated the restructuring of the atomic structure after the one atom detachment; the following maxima, smaller in magnitude, showed further minor changes in the structure. Another important energy characteristic of the nuclear system response process to external influences is repulsive energy. The profile of its change is shown in Figure 3e, during the first 40 fs against the background of a laser pulse. This graph well confirmed the previous conclusion: the mechanism of atom detachment began at ~13 fs. Primarily, there was an increase in the repulsive energy of the pulse, then a sharp decrease. The next noticeable maximum of the kinetic energy of the nuclei was observed during the restructuring of the atomic system after the atom separation.

Based on the data obtained, the response of surface fragments of the quasi-2D layer surface to incident laser radiation with a 1064 nm wavelength was studied. The purpose of this in silico study was to determine whether and to what extent fragments of graphene nanoflake would be destroyed under the influence of laser radiation. Figure 4 shows two cases of a layer fragment before and after laser exposure. In the first case, the atomic structure of the surface was characterized by a branched structure with graphene nanoflakes clearly visible above the surface of the layer (Figure 4a). It is seen that after laser exposure, some of the atoms and some small fragments broke away from the original structure. The dynamics of the change in the number of non-hexagonal elements of the graphene structure after laser exposure in the second case (Figure 4b) is also shown. Prior to the application of the laser pulse, the atomic grid contained 1 octagon, 11 pentagons, and 5 heptagons. After the exposure, the number of pentagons and heptagons decreased by one. Based on the results of a large number of numerical experiments, it was found that graphene nanoflakes, which were clearly visible above the surface of the film, did not collapse, but only individual edge atoms of graphene fragments were torn off.

Another important point of the in silico research was to identify the influence of strong electric fields on the surface of graphene and the nanotubes layer. For this purpose, an atomistic model of a supercell layer of graphene and nanotubes containing 4098 atoms was constructed. There were 4 SWCNT supercells with chirality (6,5). A hybrid of graphene nanoscale covalently bonded to a short nanotube was placed in one of the cavities. This hybrid contained 714 atoms and was not covalently bound to the environment. This predetermined the freedom of the hybrid under external influence. The junction of the graphene flake with the nanotube was formed by nonhexagonal elements: 11 pentagons and 16 heptagons (in Figure 5a, the elements are marked in red and blue, respectively). Next, an electric field was applied in the direction (−1,0,0), as shown in Figure 5a. Molecular dynamic modeling established the nature of the behavior of such a structure to an electric field with a strength of F = 0.1 V/Å. As a result of the electron charge density redistribution, the structure received a dipole moment, which caused the structure to be positioned so that the dipole moment vector was oriented along the electric lines of the external field. The time during which this happened was 5.5 ps (under these conditions). The induced magnitude of the electric dipole moment was 2424.8 Debye. Thus, as a result of the external field influence, the formation of new emitting centers was likely. The partial destruction of the edge areas can also be seen. Figure 5b shows the distribution of the electron charge density at the initial moment and at the moment when the dipole moment lines up along the force line of the external electric field. The view of the layer with the unfolded hybrid is shown in Figure 5c (highlighted in green).

Thus, pulsed laser exposure can contribute to the formation of a surface from graphene flakes and from hybrids based on graphene and SWCNT, which is accompanied by partial destruction of the edge fragments of graphene.

### 3.2. Deposition of Carbon Nanomaterials on Si Substrate

Initially prepared liquid dispersions of 1, 2, and 3 carbon nanomaterials (SWCNT, rGO, rGO + SWCNT) were deposited by spray deposition method on Si substrates, forming the corresponding layers. Despite the homogeneity of the dispersion, the formed layers on the substrate had a significant difference in height. This was clearly seen in SEM images for the rGO + SWCNT layer (Figure 6a,b). It is a well-known fact that agglomerates are present in liquid dispersions with carbon nanotubes and graphene, which are formed under the action of Van der Waals forces [104,105]. The morphology of the layer consisted of SWCNT with inclusions of large rGO particles, which led to the formation of an inhomogeneous layer in height with a spread of more than 10 microns. The inhomogeneity of the emission centers distribution along the height will lead to inhomogeneity of the emission current along the surface and, as a consequence, to rapid degradation of the cathode due to the destruction of individual centers.

Since the formation of homogeneous layers of carbon nanomaterials was hindered by the presence of large SWCNT and rGO agglomerates, the dispersions were purified from them by centrifugation. After the process of decanting large agglomerates, the height difference of the layers from all dispersions was minimal and calculated in nanometers; this was demonstrated in SEM images of layers from dispersion 1 based on SWCNT (Figure 6c,d), dispersion 2 based on rGO (Figure 6e,f), and dispersion 3 based on rGO and SWCNT (Figure 6g,h).

### 3.3. Laser Formation of Hybrid Nanostructures Based on SWCNT and rGO on Si Substrate

After the formation of homogeneous layers, the surface of the silicon chip was divided into four square areas, which were exposed to laser radiation in the scanning mode. Initially, the first method of laser processing in the Ar gas environment was applied. The external view of SWCNT, rGO, and rGO + SWCNT layer samples after laser exposure with an energy density of 0, 0.3, 0.5, and 0.8 J/cm^2^ is shown in Figure 7a–c. It can be seen from the figure that with an increase in the energy density of laser radiation, a color change occurred, which was caused by oxidation and a corresponding change in the morphology of carbon nanomaterials. This indicated that it was impossible to completely replace the atmosphere with an inert gas even in a specialized chamber; during laser exposure, a significant amount of oxygen and water was present in the chamber and nanomaterials. This fact was confirmed by SEM images of the surfaces of SWCNT (Figure 7d), rGO (Figure 7e), and rGO + SWCNT (Figure 7f) samples obtained after exposure to laser radiation with an energy density of 0.5 J/cm^2^ in an Ar gas environment. The presented results demonstrate the importance of controlling the concentration of oxygen and water vapor when exposed to pulsed laser radiation on layers of carbon nanomaterials in order to prevent damage to the latter. To minimize the presence of oxygen and water vapor, the samples were then subjected to laser exposure in a vacuum chamber.

In accordance with the modeling results, it was found that laser radiation induced the destruction of carbon bonds in carbon nanomaterials—the detachment of atoms at the edges, the formation of new bonds between nanotubes, graphene flakes, and their hybrids—due to the formation of nonhexagonal carbon rings. In addition, an external electric field affected the orientation of the dipole moment vector of hybrids as a result of the redistribution of the electron charge density of the structure. Therefore, it is necessary to experimentally determine how the morphology of nanostructures based on SWCNT, rGO, and rGO + SWCNT can change under the influence of laser radiation. For this purpose, three of the four regions on the surfaces of samples from dispersions 1, 2, and 3 were exposed to radiation energy densities of 0.3, 0.5, and 0.8 J/cm^2^. These values were chosen as extreme ones that clearly demonstrate the degree of influence of a given energy density value. In the case of exposure to laser radiation with a threshold energy density of 0.5 J/cm^2^ on the SWCNT layer, the effect of SWCNT orientation at a certain angle relative to the Si substrate was obtained in combination with minor local damage to the SWCNT at the vertices (Figure 8a,b). Exposure to laser radiation with an energy density of 0.3 J/cm^2^ did not significantly change the surface morphology of the SWCNT layer, and the energy density of 0.8 J/cm^2^ led to severe damage to the layer.

When laser exposure was applied to an rGO + SWCNT layer, the effect of hybrid rGO/SWCNT nanostructures formation was observed. SWCNT were welded to each other and to rGO flakes, forming strong bonds (Figure 8c,d). In addition to the binding of carbon nanomaterials, the effect of orientation at an angle of hybrid rGO/SWCNT nanostructures relative to the substrate under the laser radiation exposure was observed. The rGO flakes with SWCNT welded to them were lifted above the substrate, with one end of the flake fragments attached to the substrate. However, such nanostructures were characterized by low density and uniformity. This may be the reason for the low emission current density. Similarly, with the SWCNT layer, a threshold value of 0.5 J/cm^2^ radiation energy density was used for orientation since lower values did not provide the orientation effect of rGO/SWCNT hybrids, and an increase in energy above the threshold value led to sublimation.

An interesting fact was that the energy density of even 0.8 J/cm^2^ did not allow changing the orientation of the rGO layer on the Si substrate. The morphology of the rGO layer after exposure to laser radiation with an energy density of 0.8 J/cm^2^ or less remained unchanged. SEM images of the layer rGO surface before (Figure 6e,f) and after (Figure 8e,f) laser exposure had a similar character to the layer surface morphology. A further increase in the energy density above the threshold value (0.8 J/cm^2^) led to the sublimation of rGO, as shown in Figure 7e. Thus, the applied layer of rGO formed a graphene surface that was unable to rise above the substrate under the influence of laser radiation due to dense packing and strong adhesion to the Si substrate.

Despite the impossibility of raising rGO particles above the substrate, the effect of “merging” rGO flakes with the surface of the Si substrate under the influence of pulsed laser radiation can be used to form effective field emission structures. Such a structure should have good adhesion and low contact resistance with the substrate. In this regard, the resulting effect of welding a layer of rGO flakes to a Si substrate under the influence of laser radiation (0.8 J/cm^2^) was used to create a buffer layer that provides a high degree of adhesion to subsequent layers of carbon nanomaterials to the substrate.

Thus, after the formation of the buffer layer from rGO, the SWCNT layer was applied as the main layer since it demonstrated the highest density of potential emission centers per unit area with a favorable aspect ratio of centers. After exposure to laser radiation with a threshold energy density of 0.5 J/cm^2^ for SWCNT, the nanotubes, on the one hand, were firmly fixed by welding to the buffer layer and, on the other hand, were oriented vertically relative to the substrate (Figure 8g,h). The resulting hybrid nanostructure rGO(bl)-SWCNT had an increased degree of uniformity and density of emission centers compared with structures from dispersions 1 and 3.

Raman spectroscopy was used to evaluate the effect of laser radiation on the defectiveness of nanostructures from SWCNT, rGO, rGO/SWCNT, and rGO(bl)-SWCNT. A distinctive feature of the Raman spectra of SWCNT is the presence of a vibrational RBM mode (100–400 cm^−1^), which does not manifest itself in any other graphene-related structures. It is widely known that its frequency is inversely proportional to the diameter of an SWCNT [106]. There are also two equally important bands in the analysis of carbon-based materials: D (~1350 cm^−1^) and G (1580–1605 cm^−1^). It is known that the G-mode appears in all sp^2^ carbon materials and is caused by the stretching of C–C bonds in the hexagonal lattice, and the D-Band arises from the defects and disorders in the carbon lattice [107]. The intensity ratio of the I_D_/I_G_ bands is widely known in the literature and is used as an indirect assessment of the defectiveness level of carbon nanomaterials.

The analysis of structural features was carried out based on the assessment of changes in the characteristic bands for SWCNT and rGO. Table 2 contains the key parameters taken from the Raman spectra. The carbon nanotubes used in this study had a semiconductor type of conductivity (splitting the G mode into two bands, Figure 9) and had a relatively low defect rate before exposure (Table 2; 0 J/cm^2^) to laser radiation. When exposed to laser radiation, the I_D_/I_G_ ratio increased with increasing power, and there was also a displacement of the D and G bands themselves, which indicated the beginning of structural destructurization (Table 2).

SWCNT samples with an increase in the energy density of laser radiation from 0.3 to 0.8 J/cm^2^ were characterized by a gradual increase in defectiveness. The frequencies of the characteristic modes were shifted towards the values for amorphous carbon, but the distinctive RBM mode was preserved at all laser radiation energies. This indicates the preservation of the overall structural features of SWCNT. A similar relationship was observed for the rGO/SWCNT sample (Table 2). However, in this case, the defect increased much more slowly, probably because the laser radiation energy accounted for, in addition to changing the morphology of SWCNT, the orientation of the rGO flakes, and the formation of bonding between SWCNT and rGO.

In the case of rGO, the relative stacking of graphene layers was random, and the defectiveness affects all the main characteristic modes, complicating the analysis [108,109]. For rGO flakes, the I_D_/I_G_ ratio gradually decreased with a higher energy density of laser radiation (from 0.3 to 0.8 J/cm^2^); however, the rate was much weaker compared with that for pure nanotubes. At the same time, the width and the intensity of the D and G bands also slowly lowered. This was probably due to the welding of graphene flakes to Si substrate and, as a result, thinning of the layer itself.

At the same time, the rGO(bl)-SWCNT sample combined, on the one hand, the regularity of increasing the defectiveness of SWCNT with an increase in the energy of laser exposure; on the other hand, the regularity of decreasing the defectiveness of rGO with an increase in the energy density of laser exposure. As a result, the initial defectiveness of this sample was higher than the defectiveness of SWCNT but lower than the defectiveness of rGO, and with an increase in energy density, the defectiveness of the rGO(bl)-SWCNT nanostructure increased, but not as rapidly as for SWCNT. A possible reason for such a defect change was the stabilization of SWCNT using an rGO buffer layer welded to the Si substrate.

### 3.4. Influence of Laser Radiation on Mechanical Properties of Hybrid Nanostructures

To determine the strength of the nanostructures formed by laser radiation, the hardness of the formed layers on the Si substrate was measured using the nanoindentation method. The samples’ hardness could change due to various patterns of carbon nanomaterials binding to each other, including the formation of rGO and SWCNT hybrids. For each sample, five measurements were carried out, and then the average hardness values were calculated. Based on the data obtained, a graph of the layers’ hardness before and after laser exposure was plotted (Figure 10).

The SWCNT layer had a hardness of 9.2 GPa. After laser exposure, the hardness increased by more than 2.5 times and amounted to 24.5 GPa. However, the hardness of the layer of rGO flakes at 19.6 GPa was initially higher than that of the SWCNT layer, while laser exposure made it possible to increase the hardness by more than 2.3 times. The layer formed from rGO and SWCNT flakes had the lowest hardness, ~7.8 GPa, but laser radiation made it possible to increase it by 2.2 times. The maximum hardness was achieved for the rGO(bl)-SWCNT nanostructure based on the rGO buffer layer and the main SWCNT layer, 22.6 and 54.4 GPa before and after laser exposure, respectively. On the one hand, the obtained patterns of increase in hardness confirmed the effect of laser welding of carbon nanomaterials to each other under the action of pulsed laser radiation. On the other hand, the samples in which a densely packed layer of rGO flakes was formed before and after laser exposure (a layer of rGO and rGO(bl)-SWCNT) had a higher hardness compared with the branched layer structure of SWCNT and rGO/SWCNT.

In addition, to control the hardness of the layers, it was important to determine the adhesion degree of the layers to the Si substrate since the vertical electrical conductivity from the bottom of the substrate to the tip of the formed nanostructure played a decisive role in the creation of stable field emission cathodes [110,111]. In this regard, a sclerometric of the layers was carried out. It was found that for the formation of scratches with identical geometric dimensions, it was necessary to apply different force values, which ensured the detachment of the layer from the Si substrate. The force values for the separation of all layers varied in the range of 7.8–10.5 mN: SWCNT (9.5 mN), rGO (10.9 mN), rGO/SWCNT (7.8 mN), and rGO(bl)-SWCNT (10.5 mN). After laser exposure, the force values increased by more than 3 times—31.8–47.5 mN. Layers based on rGO and rGO(bl)-SWCNT had the maximum adhesion after laser exposure. The force for detachment of these layers was 44.8 and 47.5 mN, respectively. The force for the SWCNT-based layer was 32.9 mN. In this case, the sample with a branched surface of raised graphene flakes was characterized by the lowest adhesion to the surface of the Si substrate. The force for this sample was 18.5 mN.

Thus, the formed rGO(bl)-SWCNT hybrid nanostructure, because of the presence of the rGO buffer layer, had a high hardness value and a high degree of adhesion to the Si substrate, which would potentially help to reduce the contact resistance and could increase the long-term stability of the field emitters characteristics based on them.

### 3.5. Effect of Laser Radiation on the Emission Characteristics of Hybrid Nanostructures

After comparing the structural and mechanical properties, the field emission current–voltage characteristics of laser-formed layers from carbon nanostructures SWCNT (Figure 11a), rGO (Figure 11b), rGO/SWCNT (Figure 11c), and rGO(bl)-SWCNT (Figure 11d) were obtained. For the rGO sample, the maximum current value was not more than 1.7 nA. Such a low value was due to the rGO layer morphology, which was characterized by the absence of surface branching and the absence of pronounced field emission centers (Figure 8e,f). Despite the attractive rGO/SWCNT structure in terms of field emission, a low field emission current was obtained (Figure 11c). This was due to the low degree of uniformity and low population density of emission centers on the layer surface (Figure 8c,d). At the same time, strong fluctuations were presented in the current–voltage characteristic, which was probably associated with low mechanical characteristics of vertically oriented graphene flakes with SWCNT relative to the Si substrate (Figure 8c). Meanwhile, for the SWCNT sample (Figure 11a), the current–voltage characteristic demonstrated high values of the maximum current ~0.14 mA. This effect was most likely due to two reasons. The first reason was the high population density of field emission centers with a high aspect ratio. The second reason was related to the effect of laser welding of the SWCNT to the silicon substrate, which reduced the contact resistance, thus improving the electron transport [92]. However, the current–voltage curve of the SWCNT layer contained significant fluctuations in the emission current with increasing voltage above 85 V, which could be explained by insufficient strength and adhesion of the SWCNT layer to the Si substrate. In order to improve the adhesion of SWCNT to the Si substrate, an rGO buffer layer was formed before the main SWCNT layer. This made it possible to improve electron transfer from the Si substrate through the rGO buffer layer to the SWCNT emitting layer. It is assumed that the covalent bonding of SWCNT to the graphene flakes may contribute to an increase in the vertical electrical conductivity [112]. For the rGO(bl)-SWCNT hybrid nanostructure sample, an increase in voltage up to 140 V demonstrated a low level of emission current fluctuations (Figure 11d). This also indicated that laser exposure of the rGO buffer layer promoted welding of graphene flakes with Si substrate, thereby increasing the field emission stability.

The important point was to ensure that the structure morphology affects the field emission characteristics. Since the morphology was controlled by varying the energy density of pulsed laser radiation, the emission current–voltage characteristics were measured using the example of an rGO(bl)-SWCNT layer formed with a radiation energy density of 0.3, 0.5, and 0.8 J/cm^2^. Figure 12 shows the average values of the emission current at 5 points of each square area formed by a given radiation energy density for the rGO(bl)-SWCNT sample. The highest field emission current was 175–226 μA at a laser radiation energy density of 0.5 J/cm^2^. Further increase in the laser radiation energy density up to 0.8 J/cm^2^ led to a decrease in the maximum emission current values below 60 μA, which could be explained by defects formation on the SWCNT surface and by the destruction of individual SWCNTs, which were effective emitters. An energy density of 0.3 J/cm^2^ was insufficient for the SWCNT orientation and formation of emission centers. The result obtained correlated with the data obtained by Raman spectroscopy.

To determine the integral emission current density, the round-shaped cathode samples of ~1 mm^2^ (Figure 13a,b) were made of SWCNT and rGO(bl)-SWCNT layers. For this purpose, the corresponding dispersions were deposited through a stencil mask with a hole of 1.1 mm in diameter. Before measuring the field emission current–voltage characteristics of the SWCNT and rGO(bl)-SWCNT, the samples were trained at a fixed emission current of 1 mA for at least 540 min. The voltage–time curves at a constant emission current of 1 mA for the SWCNT and rGO(bl)-SWCNT samples are shown in Figure 13c,d, respectively. For the SWCNT sample, a significant increase in voltage (more than 100 V) was observed, which indicated sample degradation during field emission due to the mechanical instability of the SWCNT on the Si substrate. For the rGO(bl)-SWCNT sample, a smooth voltage increase followed by stabilization was obtained. This indicated an improved adhesion of SWCNT to the substrate compared with SWCNT due to laser formation of the buffer layer. This fact was confirmed by the measured current–voltage characteristics after training of layers based on SWCNT and rGO(bl)-SWCNT hybrids (Figure 13e). Then for the SWCNT sample, the maximum current was 2.5 times less than that of the rGO(bl)-SWCNT sample at the same field strength. The current density for the SWCNT sample was 226 mA/cm^2^, and for the rGO(bl)-SWCNT—562 mA/cm^2^.

We compared the obtained values of the emission current density with the values for structures based on carbon materials reported by other researchers. Based on the analysis, it was found that obtained emission current density values of rGO(bl)-SWCNT hybrid sample 562 mA/cm^2^ exceed the values obtained earlier. Murakami et al. obtained the emission current density of the GO structure at 100 mA/cm^2^ [40]. Kaur et al., in their work, obtained the value of the hybrid CNT/rGO structure’s current density of 64 mA/cm^2^ [42]. In work [56], emission current density of CNT film up to 1 mA/cm^2^ was achieved.

## 4. Conclusions

Technology for the formation of hybrid nanostructures by pulsed 1064 nm laser radiation on spray-deposited layers of homogeneous liquid dispersions of single-walled carbon nanotubes (SWCNT), reduced graphene oxide (rGO), and SWCNT with rGO on a Si substrate was developed.

Molecular dynamics modeling using the real-time time-dependent density functional tight-binding (TD-DFTB) method showed that laser radiation rearranged SWCNT with graphene flakes into hybrid nanostructures with the formation of nonhexagonal graphene elements. Because of the electron charge density redistribution, the hybrid structure acquired a dipole moment oriented along the external field and was located along it. SEM images confirmed the effect of changing the SWCNT and rGO/SWCNT orientation upon pulsed laser radiation exposure with an energy density of 0.5 J/cm^2^.

It was found that rGO/SWCNT nanostructures had lower hardness and adhesion to the Si substrate compared with rGO and were less efficient as field emitters compared with SWCNT due to the low uniformity and density of emission centers. Therefore, hybrid nanostructures rGO(bl)-SWCNT were proposed with a laser-exposed rGO buffer layer (0.8 J/cm^2^) and a main layer of SWCNT (0.5 J/cm^2^). The laser provided SWCNT welding to graphene and changed their orientation. With the help of Raman spectroscopy, the effect of laser radiation energy on the defectiveness of all nanostructures of carbon nanomaterials was determined.

The laser exposure provided an increase in the hardness of all nanostructures more than twice. The maximum hardness obtained for rGO(bl)-SWCNT was 54.4 GPa. In addition, the adhesion of rGO to the substrate and electron transport between the substrate and rGO(bl)-SWCNT increased. The rGO(bl)-SWCNT cathode with an area of ~1 mm^2^ provided stable field emission for 540 min at a current of 1 mA. The achieved current density was 2.5 times higher in comparison with SWCNT cathode and amounted to 562 mA/cm^2^.

The proposed technology for creating hybrids from SWCNT and rGO has great potential for developing emission cathodes for electronic devices, such as X-ray tubes, field emission displays, and vacuum microwave devices.

## Figures and Tables

**Figure 1 nanomaterials-12-02812-f001:**
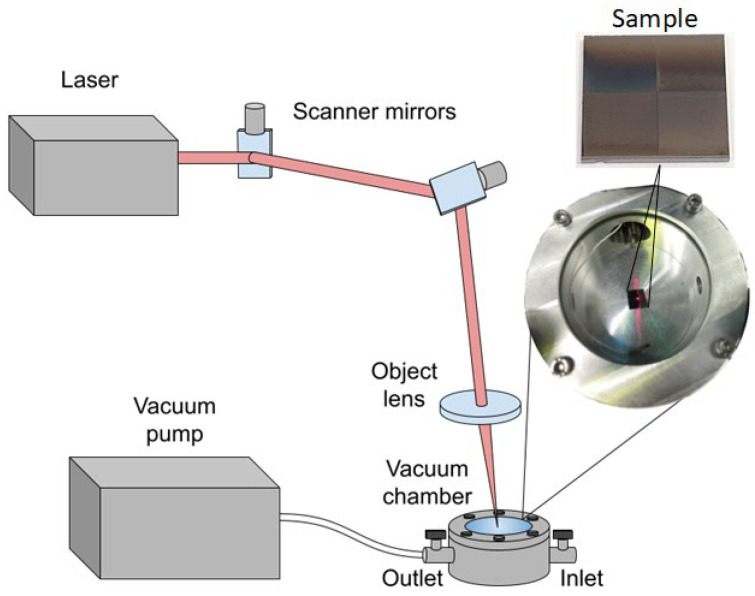
Scheme of laser setup for the formation of carbon nanomaterials layers.

**Figure 2 nanomaterials-12-02812-f002:**
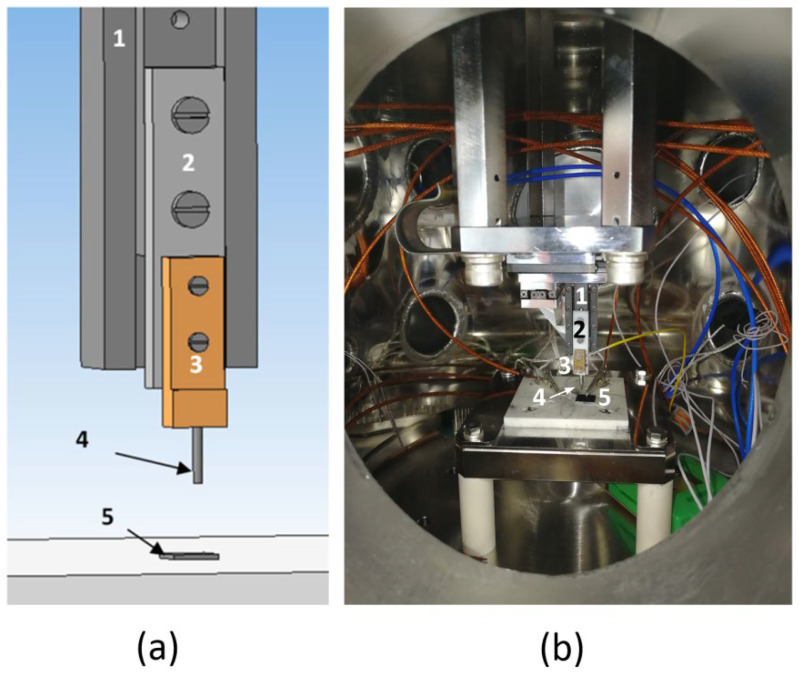
Experimental setup for measuring field emission, containing an anode positioning system relative to the sample in a vacuum chamber (1—SLC-17, 2—fluoroplastic plate, 3—contact plate, 4—anode, 5—sample): (**a**) Scheme and (**b**) External view.

**Figure 3 nanomaterials-12-02812-f003:**
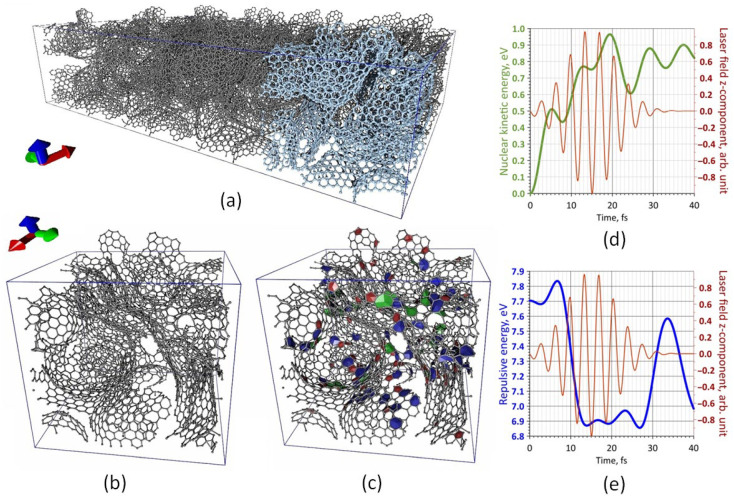
Atomic structure of a supercell of graphene nanoflakes layer: (**a**) General view of a layer fragment, (**b**,**c**) Two types of supercells; energy characteristics of the carbon atom detachment process under the action of the laser pulse: (**d**) Kinetic energy of the nuclei, (**e**) Repulsive energy.

**Figure 4 nanomaterials-12-02812-f004:**
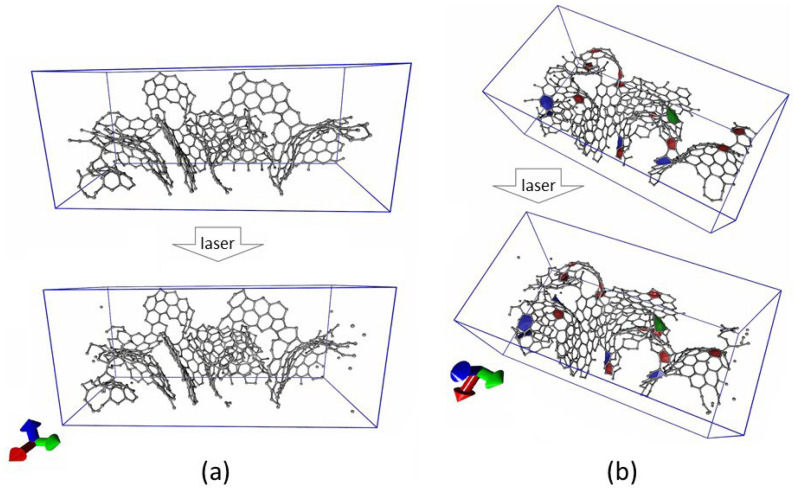
Atomistic structure of the quasi-2D layer surface: (**a**) Before and (**b**) After laser exposure.

**Figure 5 nanomaterials-12-02812-f005:**
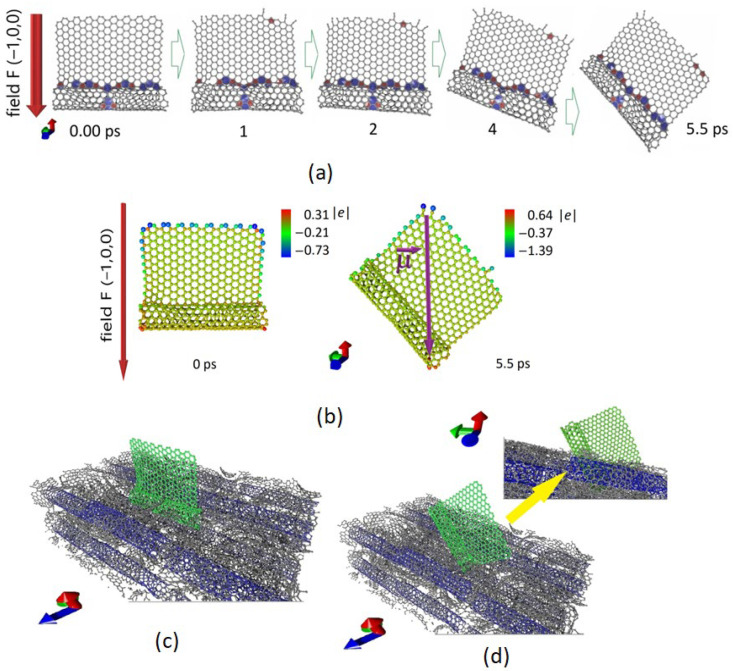
(**a**,**b**) Response of a hybrid based on a graphene flake bonded to SWCNT to an external electric field over time; a fragment of the layer surface of hybrids based on graphene flakes bonded to SWCNT, (**c**) before, and (**d**) after laser exposure.

**Figure 6 nanomaterials-12-02812-f006:**
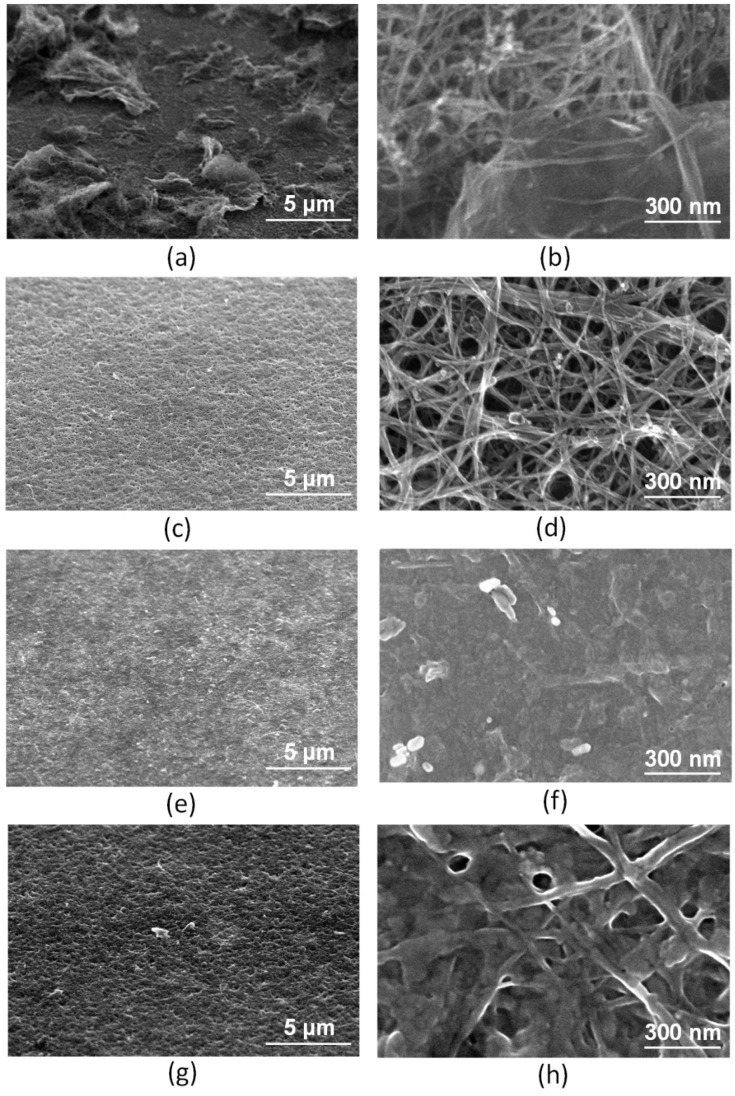
SEM image of the deposited layer of rGO + SWCNT: (**a**) top view and (**b**) view at 52° angle; (**c**,**d**) after centrifugation of SWCNT layers, (**e**,**f**) rGO, (**g**,**h**) rGO + SWCNT.

**Figure 7 nanomaterials-12-02812-f007:**
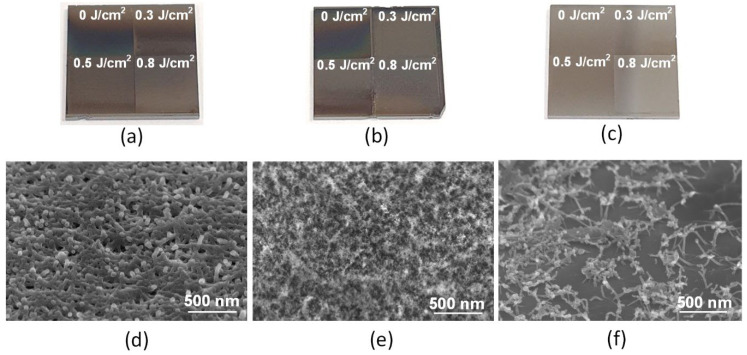
External view of (**a**) SWCNT, (**b**) rGO, and (**c**) rGO + SWCNT layers after laser exposure with energy densities of 0, 0.3, 0.5, and 0.8 J/cm^2^ in an Ar gas environment; (**d**–**f**) Corresponding SEM images obtained when exposed to an energy density of 0.5 J/cm^2^.

**Figure 8 nanomaterials-12-02812-f008:**
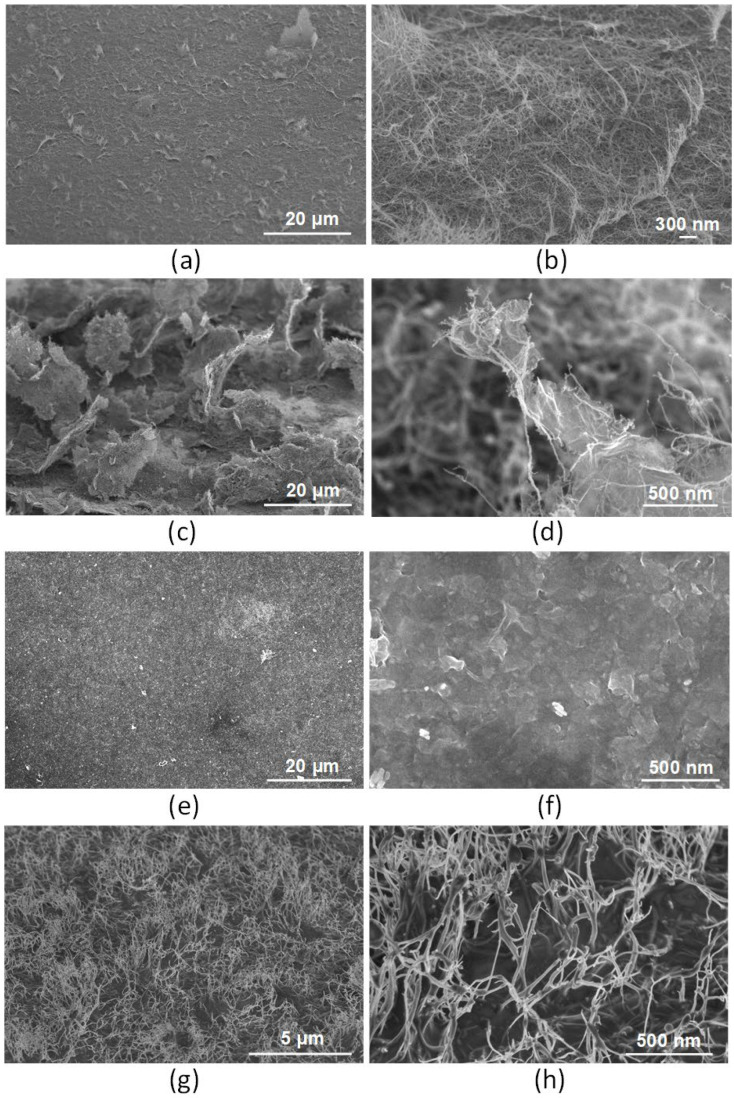
SEM images of layer structures based on: (**a**,**b**) SWCNT, (**c**,**d**) rGO/SWCNT hybrids after laser exposure with an energy density of 0.5 J/cm^2^; (**e**,**f**) rGO layer after laser exposure with an energy density of 0.8 J/cm^2^, and (**g**,**h**) rGO(bl)-SWCNT hybrid nanostructure with a buffer layer of rGO after laser exposure with an energy density of 0.8 J/cm^2^ and a main layer of SWCNT after laser exposure with an energy density of 0.5 J/cm^2^.

**Figure 9 nanomaterials-12-02812-f009:**
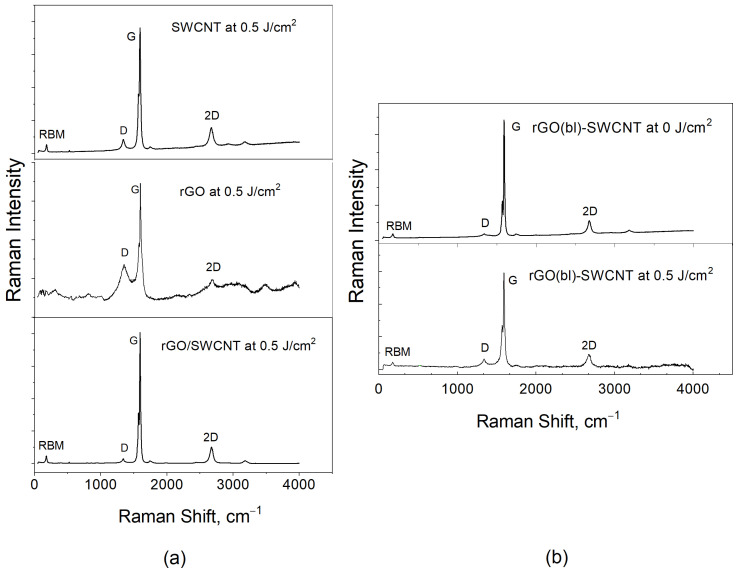
(**a**) Raman spectra of SWCNT, rGO, and rGO/SWCNT samples after laser exposure with an energy density of 0.5 J/cm^2^ and (**b**) Raman spectra of rGO(bl)-SWCNT before and after laser exposure with an energy density of 0.5 J/cm^2^.

**Figure 10 nanomaterials-12-02812-f010:**
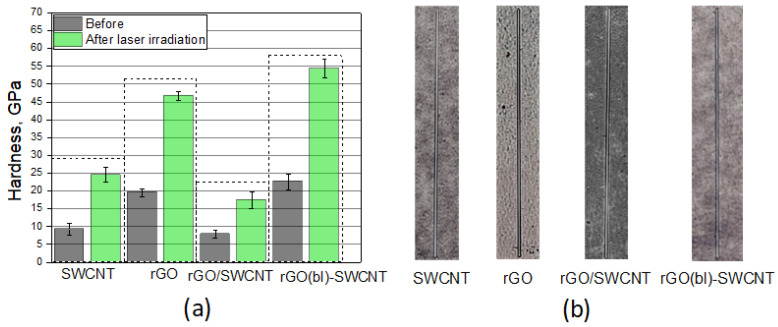
(**a**) Hardness before and after laser exposure and (**b**) Scratches formed on SWCNT, rGO, rGO/SWCNT, and rGO(bl)-SWCNT layers after laser exposure.

**Figure 11 nanomaterials-12-02812-f011:**
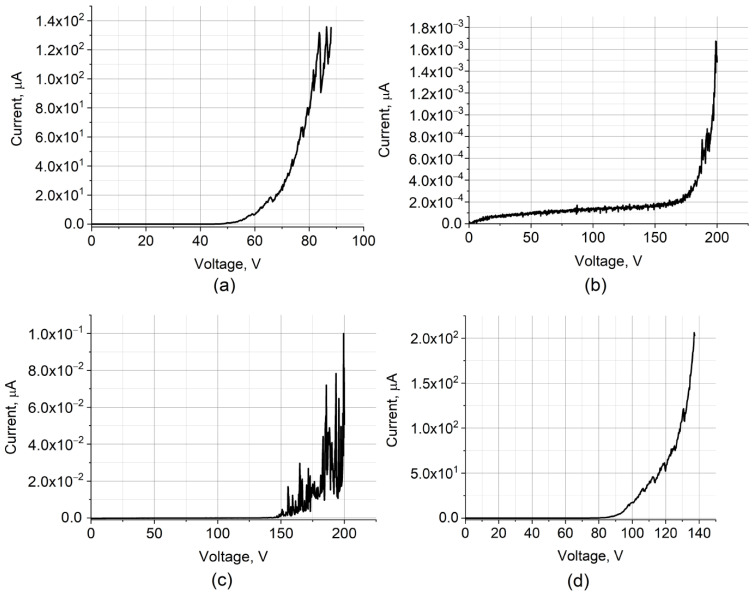
Current–voltage characteristics of layers based on: (**a**) SWCNT, (**b**) rGO, (**c**) Hybrid nanostructures rGO/SWCNT, and (**d**) rGO(bl)-SWCNT.

**Figure 12 nanomaterials-12-02812-f012:**
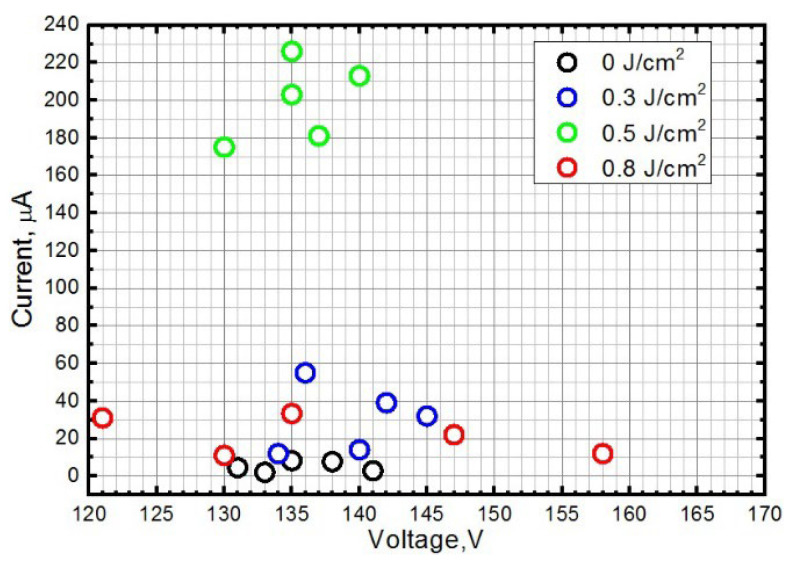
The maximum current–voltage values of the layer based on rGO(bl)-SWCNT for the region before (0 J/cm^2^) and after laser exposure with an energy density of 0.3, 0.5, and 0.8 J/cm^2^.

**Figure 13 nanomaterials-12-02812-f013:**
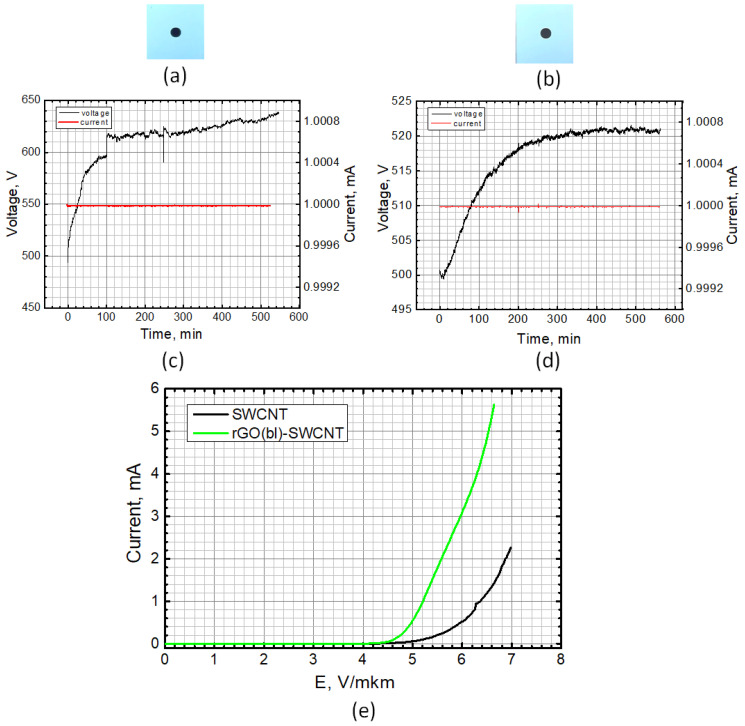
External view of a field emission cathode based on: (**a**) SWCNT and (**b**) rGO(bl)-SWCNT hybrid nanostructures, corresponding to the voltage–time characteristic at a constant emission current of 1 mA for (**c**) SWCNT and (**d**) rGO(bl)-SWCNT cathodes; (**e**) Current–voltage characteristics for SWCNT (black line) and rGO(bl)-SWCNT (green line) cathodes.

**Table 1 nanomaterials-12-02812-t001:** Dispersions compositions.

Dispersion Number	Composition	Concentration (mg/mL)
1	SWCNT	0.1
2	rGO	0.1
3	rGO + SWCNT	0.05/0.05

**Table 2 nanomaterials-12-02812-t002:** Characterization of Raman spectra via characteristic bands for SWCNT, rGO, rGO/SWCNT, and rGO(bl)-SWCNT nanostructures.

Sample	Laser Energy Density, J/cm^2^	I_D_/I_G_	ν (G Band), cm^−^^1^	ν (D Band), cm^−^^1^
1. SWCNT	0	0.036	1595	1343
	0.3	0.083	1592	1338
	0.5	0.112	1590	1338
	0.8	0.269	1589	1339
2. rGO	0	0.439	1596	1352
	0.3	0.425	1597	1354
	0.5	0.421	1597	1353
	0.8	0.407	1596	1353
3. rGO/SWCNT	0	0.032	1595	1343
	0.3	0.033	1593	1341
	0.5	0.038	1593	1340
	0.8	0.054	1591	1339
4. rGO(bl)-SWCNT	0	0.092	1594	1340
	0.3	0.113	1595	1339
	0.5	0.136	1589	1341
	0.8	0.181	1592	1342

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
