# Peer review of "Hybrid Carbon Nanotubes–Graphene Nanostructures: Modeling, Formation, Characterization"

_nanomaterials, 2022, doi:10.3390/nano12162812_

Round 1

Reviewer 1 Report

 The Manuscript ID: nanomaterials-1868793 describes an interesting study regarding the preparation and characterization of carbon nanomaterials. 

I made myself some suggestions/corrections in the attached manuscript. Authors must pay attention to the green highlighted words/sentences. Briefly, I suggested the following:

1) Moderate English changes required.

2) Lines 32-33: Please insert the abbreviation in brackets: "buffer layer (bl)".

3) Line 175: The abbreviation must be detailed: "Self-consistent field (SCF)".

4) Lines 187-190; 201-203; 211 (Table 1): Replace “ml” with “mL”.

5) Line 192: Please give the name of centrifuge. Please insert a space between the numerical value and the measurement unit: "12000 g".

6) Line 193: The authors must specify in the text what type of water they used.

7) Line 208: Please insert a space between the numerical value and the measurement unit: "20000 g".

8) Line 222: The standard deviation is too high (500±100 nm). Please verify.

9) In the section "2.3. Method of Laser Formation of Hybrid Nanostructures", the authors must describe the preparation of  rGO(bl)-SWCNT.

10) Line 248: Replace "above" with "below".

11) Line 260: Write "etc."

12) Line 275: Replace: "subsrates" with "substrates".

13) Line 319: Write: "layer of ~4.2 nm thick"

14) Lines 335-337: Reformulate the sentence!

15) Line 386: Write: "of the in silico research".

16) Lines 434-436: Figure 6:  In both cases, the SEM images with both magnifications (5 µm and 300 nm) must be inserted.

17) Line 516: Attention to the abbreviation "bl"! All the abbreviations must be detailed at their first appearance in the text.

18) Line 595: Replace "exosure" with "exposure".

19) Line 611: Please correct the green highlighted afirmation and correct the maximum current value.

20) Lines 648-649:  Revise the the green highlighted sentence!!! The highest field emission current was in the range of 160–240 µA!  Verify!

21) Line 662: Insert "of" in  "hole of 1.1 mm".

22) Lines 824-827: Complete the page numbers in the references [40] and [41].

Reviewer 2 Report

Submitted research article as “Hybrid carbon nanotubes-graphene nanostructures: modeling, formation, characterization" in interesting and after carefully observation, it was found that this article needs major revision because several issues and explanations are still need to be clarified. I recommend it publication in this journal (MAJOR REVISION) after providing proper improvement in revised version by including suggestion, modification and reply to raised queries which are given below.

1.      What is the main advantage of this reported research work and how  it is different from other research work?

2.      Try to differentiate this work with previous publication based on porous graphene oxide including synthesis approach, cost-effective and valuable materials.

3.      I cannot find a real literature review in introduction section and why authors have done this work, what is the scientific gap, and what is novelty in this submitted research work.

4.      Introduction section is not well written and organized by relevant articles especially published on graphene derivatives based composites. Improve the introduction section by including the suggested references as Critical Reviews in Solid State and Materials Sciences 46 (5), 385-449, 2021; ACS Applied Nano Materials 2 (7), 4626-4636, 2019; Journal of Energy Storage 40, 102724, 2021; Applied Surface Science 416, 259-265, 2017; Nanoscale 14 (25), 8914-8918, 2022;

5.      Check the wt% of C and O in reduced graphene oxide materials. How these elemental contents of C and O in reduced graphene oxide affect the properties?

6.      In Figure 7, how laser exposure form porosity on its surfaces as seen in figures 7.

7.      What was the effect of laser frequency (Progress in Energy and Combustion Science 91, 100981, 2022) on laser exposure? Try to elaborate in details in revised version with help of suggested article.

8.      How ratio of 2D band to D band determine the number of layers? Explain it for Fig. 9.

9.      There are some grammatical and punctuation errors in this manuscript. The English language should be improved. Tenses are not consistent from sentence to sentence and there are some grammatical errors.
